# A Real-World Application of Liquid Biopsy in Metastatic Colorectal Cancer: The Poseidon Study

**DOI:** 10.3390/cancers13205128

**Published:** 2021-10-13

**Authors:** Letizia Procaccio, Francesca Bergamo, Francesca Daniel, Cosimo Rasola, Giada Munari, Paola Biason, Stefania Crucitta, Giulia Barsotti, Giulia Zanella, Valentina Angerilli, Cristina Magro, Silvia Paccagnella, Veronica Di Antonio, Fotios Loupakis, Romano Danesi, Vittorina Zagonel, Marzia Del Re, Sara Lonardi, Matteo Fassan

**Affiliations:** 1Oncology Unit 1, Department of Oncology, Veneto Institute of Oncology—IRCCS, 35128 Padova, Italy; letizia.procaccio@iov.veneto.it (L.P.); francesca.bergamo@iov.veneto.it (F.B.); francesca.daniel@iov.veneto.it (F.D.); cosimo.rasola@iov.veneto.it (C.R.); paola.biason@iov.veneto.it (P.B.); giulia.barsotti@iov.veneto.it (G.B.); giulia.zanella@iov.veneto.it (G.Z.); cristina.magro@iov.veneto.it (C.M.); veronica.diantonio@iov.veneto.it (V.D.A.); fotiosloupakis@gmail.com (F.L.); vittorina.zagonel@iov.veneto.it (V.Z.); 2Department of Surgery, Oncology, and Gastroenterology, University of Padova, 35121 Padova, Italy; 3Surgical Pathology Unit, Department of Medicine (DIMED), University of Padova, 35121 Padova, Italy; giada.munari@gmail.com (G.M.); valentina.angerilli@gmail.com (V.A.); silviapak80@gmail.com (S.P.); matteo.fassan@gmail.com (M.F.); 4Veneto Institute of Oncology (IOV-IRCCS), 35128 Padova, Italy; 5Clinical Pharmacology and Pharmacogenetics Unit, Department of Clinical and Experimental Medicine, University Hospital of Pisa, 56121 Pisa, Italy; stefania.crucitta@gmail.com (S.C.); romano.danesi@ao-pisa.toscana.it (R.D.); marzia.delre@gmail.com (M.D.R.); 6Oncology Unit 3, Department of Oncology, Veneto Institute of Oncology—IRCCS, 35128 Padova, Italy

**Keywords:** liquid biopsy, metastatic colorectal cancer, real-world, first-line therapy, concordance, accuracy, turnaround time

## Abstract

**Simple Summary:**

The choice of first-line regimen represents a critical step in the therapeutic road for patients with metastatic colorectal cancer (mCRC) because the response to first-line treatment is the primary determinant of outcomes. The extended characterization of *RAS* and *BRAF* mutational status is one the key points to select the optimal first-line therapy. Despite the great strides, there are still many challenges in the decision-making regarding the upfront strategy in mCRC patients. *RAS* testing turnaround time and the unavailability of tissue are the factors most frequently cited by physicians for treating mCRC patients with unknown *RAS* status, particularly in small volume centers. An accurate blood-based *RAS/BRAF* mutation assay with fast turnaround time would help circumvent these hurdles. Our experience suggests that genotyping *RAS/BRAF* to guide therapy may be the first use of liquid biopsy in the daily care with mCRC patients, particularly in the urgent situation to define the first-line regimen.

**Abstract:**

Background: First-line decision making is the key to the successful care of mCRC patients and *RAS/BRAF* status is crucial to select the best targeted agent. In hub centers, a relevant proportion of patients referred from small volume centers may not have standard tissue-based (STB) molecular results available at the time of the first visit (T0). Liquid biopsy (LB) may help circumvent these hurdles. Methods: A monoinstitutional prospective head-to-head comparison of LB versus (vs.) STB testing was performed in a real-world setting. Selection criteria included: mCRC diagnosis with unknown *RAS/BRAF* status at T0, tumoral tissue archived in external centers, no previous treatment with anti-EGFR. At T0, patients underwent plasma sampling for LB testing and procedure for tissue recovery. *RAS/BRAF* genotyping was carried out by droplet digital PCR on circulating-tumoral (ct) DNA. The primary endpoint was the comparison of time to LB (T1) vs. STB (T2) results using the Mann–Whitney U test. Secondary endpoints were the concordance between LB and STB defined as overall percent agreement and the accuracy of LB in terms of specificity, sensitivity, positive and negative predictive value. We also performed an exploratory analysis on urinary (u) ctDNA. Results: A total of 33 mCRC patients were included. Mean T1 and T2 was 7 and 22 days (d), respectively (*p* < 0.00001). T2 included a mean time for archival tissue recovery of 17 d. The overall percent agreement between LB and STB analysis was 83%. Compared to STB testing, LB specificity and sensitivity were 90% and 80%, respectively, with a positive predictive value of 94% and negative one of 69%. In detail, at STB and LB testing, *RAS* mutation was found in 45% and 42% of patients, respectively; *BRAF* mutation in 15%. LB results included one false positive and four false negative. False negative cases showed a significantly lower tumor burden at basal CT scan. Concordance between STB and uctDNA testing was 89%. Conclusions: Faster turnaround time, high concordance and accuracy are three key points supporting the adoption of LB in routinary mCRC care, in particular when decision on first-line therapy is urgent and tissue recovery from external centers may require a long time. Results should be interpreted with caution in LB wild-type cases with low tumor burden.

## 1. Introduction

A key point in the successful management of patients with metastatic colorectal cancer (mCRC) is first-line treatment decision making. According to major guidelines and experts’ recommendations, therapeutic choice could be guided by a combination of clinical considerations (i.e., patient’s age and performance status, comorbidities, expectations and preferences, primary tumor location, treatment toxicity profile, physician’s experience) and tumor molecular characteristics (i.e., *RAS* and *BRAF* mutational status) [1]. New emerging molecular markers such as microsatellite instability (MSI) status or mismatch repair (MMR) proteins expression, *HER2* amplification or *NTRK* rearrangements are progressively entering the list of genes with relevant clinical implications [2,3,4,5,6].

*RAS* mutational status represents a criterion for selecting patients who would benefit from anti-*EGFR* therapy because these drugs are effective [7,8] and registered in *RAS* wild-type patients only, by the US Food and Drug Administration (FDA) and European Medicines Agency (EMA). Despite similar data, *BRAF* V600E mutation has never been definitely established as negative predictive determinant to anti-EGFRs [9,10], but major guidelines recommend testing for adjusting therapeutic intensity based on its strong prognostic effect and association to peculiar clinical characteristics [11,12,13].

According to National Comprehensive Cancer Network (NCCN) and American Society of Clinical Oncology (ASCO) guidelines, the detection of *RAS* and *BRAF* mutations is routinely carried out in tumor tissue from either biopsies or resected specimens of the primary tumor or of a metastatic lesion [13].

Unlike other diseases, such as non-small-cell lung cancer (NSCLC), tumor tissue availability is not a major hurdle in the ordinary management of mCRC. As a matter of fact, around 50% of mCRC patients present with metachronous metastases and therefore a previously resected primary tumor should be available in surgical pathology archives. Even in cases with synchronous metastatic onset, colonoscopy and/or liver biopsy make the access to tumor material relatively easy.

It is well known that patients often access high volume referral centers for initial surgery or for second opinion at the time of first-line treatment decision making. This causes the common situation of the need for retrieving as soon as possible the archival samples stored elsewhere and timely execution of molecular tests to allow the hub centers to select the optimal first-line therapy in advanced stages. In order to face this or any other issue for which tumor tissue is not available (i.e., insufficient quantity or quality of archival samples), circulating tumoral (ct) DNA analysis has been proposed as possible alternative for rapid molecular testing [13,14,15,16].

Nevertheless, there are several open questions that need to be answered before adopting ctDNA testing in the daily management of mCRC patients [17,18,19,20]. Data on real advantages in terms of turnaround time and reliability of results are limited [21].

Based on the above reported observations, we designed the present prospective study with the primary objective of evaluating the turnaround time of plasma *RAS/BRAF* analyses compared to that of standard tissue-based (STB) methods in a cohort of consecutive mCRC patients referred to our center, with no molecular data at the time of their clinical oncology assessment for treatment decision making. In our exploratory longitudinal cohort observational study, we focused only on patients with advanced disease referred to our center for a second opinion because we were certain to find a greater amount of ctDNA than what it was present in patients at early stage of CRC. Secondary objectives included the accuracy of plasma-based *RAS/BRAF* testing conducted by means of digital droplet PCR (ddPCR), and exploratory analysis on the role of a fully automated technology (i.e., Idylla^TM^) [22] and on urine samples as alternative source of ctDNA (uctDNA) [23,24].

## 2. Materials and Methods

### 2.1. Study Design

The key eligibility criteria were the following: histological diagnosis of colorectal adenocarcinoma, metastatic disease, no previous testing for *RAS* and/or *BRAF* mutations, no previous treatment with anti-*EGFR* targeted monoclonal antibodies.

The primary objective was to describe the current practice regarding STB and liquid biopsy (LB)-based molecular testing for *RAS* and *BRAF* mutations as relevant markers for clinical decision making in mCRC patients. For this purpose, the primary endpoint was to determine the turnaround time of LB (T1) compared to that of STB analysis (T2) recording the following time intervals (expressed in days): between the plasma sampling and LB result (turnaround of LB analysis, TLBA), between the plasma-sample shipment to the referent analysis laboratory and LB result (turnaround of analysis laboratory, LAB-time), between the recovery request of tissue sample from the center where it was collected and the tissue *RAS/BRAF* testing result (turnaround of STB analysis, TSTBA), and between the tissue sample entry to the referent pathology unit and STB result (turnaround of STB analysis, PAT-time). Blood sampling and tissue *RAS/BRAF* mutational status testing were performed by the research nurses of Veneto Institute of Oncology and by pathology department of University of Padua, respectively. *RAS/BRAF* status in LB was tested at the Pharmacogenetic Laboratories of the University Hospital of Pisa.

Secondary endpoints were the following: concordance between STB and LB analysis, accuracy of LB in terms of sensitivity, specificity, negative and positive predictive values. Notably, to evaluate concordance of LB versus (vs.) tissue *RAS/BRAF* testing results, positive percent agreement, negative percent agreement, and overall percent agreement were calculated. To estimate the LB accuracy, we evaluated the specificity, sensitivity, positive and negative predictive value compared to STB results considered as standard reference. Among secondary endpoints, we also included exploratory analyses either on plasma *KRAS* testing by the novel fully automated Idylla^TM^ method and on uctDNA [23,24] compared to the STB genotyping in terms of turnaround times, concordance and accuracy in a real-world setting of mCRC patients.

### 2.2. Sampling and Molecular Testing

A single peripheral blood sample from each patient was collected in a K2 EDTA Vacutainer tube. Plasma samples were prepared from collected blood within 4 h of phlebotomy with two subsequently centrifugation: the first one at 1600× *g* for 10 min, then the isolated plasma was centrifuged at 3000× *g* for 10 min and stored at −80°C. All plasma samples were shipped in dry-ice to the Pharmacogenetic Laboratories at the University Hospital of Pisa, where *RAS/BRAF* mutation analysis was carried out on ctDNA by using a digital droplet PCR (ddPCR, Biorad, Hercules, CA, USA).

Among enrolled patients harboring mutations in *KRAS* exons 2, 3 or 4, we selected 4 consecutive subjects to prospectively assess the presence of these mutations in their plasma samples also by the Idylla^TM^ platform technique [23].

### 2.3. Plasma-Based RAS/BRAF Mutation Testing

CtDNA was extracted from 3 mL of plasma samples using the QIAmp circulating nucleic acid kit (Qiagen, Valencia, CA, USA). The ddPCR™ Screening Multiplex Kit (Bio-Rad, Hercules, CA, USA) were used to assess the mutational status of *KRAS* (codons 12 and 13), *NRAS* (codons 12,13 and Q16) and *BRAF* (codon V600) genes. We employed QX200 ddPCR system (Bio-Rad, Hercules, CA, USA) using specific ddPCR Supermix with no dUTTP for probes. DdPCR analysis is based on the co-amplification of mutant or wild-type alleles (FAM- and HEX-labeled probes, respectively) [25]. QX200 droplet generator and C1000 Touch Thermo Cycler (Bio-Rad, Berkeley, CA, USA) were used for the DNA amplification with the following protocol: 95 °C for 10 min followed by 40 cycles of 94 °C for 30 s and 55 °C for 1 min, then 98 °C for 10 min. DdPCR allows for the enumeration of rare mutant variants in complex mixtures of DNA (wild type and mutant) based on an emulsion droplet technology and Poisson’s distribution. Mutation specific amplification occurs in each individual droplet, and counting the positive droplets gives precise and absolute target quantification. The results, obtained using a QX200 ddPCR system (Bio-Rad, Valencia, CA, USA), are reported as copies per ml (copies/mL) of mutant DNA alleles.

Droplets were read in the QX200 droplet reader (Bio-Rad, Berkeley, CA, USA) and analyzed using the Quantasoft software version 1.0.596 (Bio-Rad, Berkeley, CA, USA). 

The software calculated the value of MAF as the ratio of drops positive for the mutant allele to drops positive for the mutant allele plus drops positive for the wild-type allele (percentage of mutant *KRAS* alleles). The sensitivity allowed the detection of very low allele frequencies down to <0.02%.

### 2.4. Tissue-Based RAS/BRAF Mutation Testing

For each patient, five 10 µm paraffin embedded sections were used to extract the DNA using the QIAmp FFPE tissue Kit (Qiagen) according to the manufacturer’s instructions. Extracted DNA was quantified using to spectrophotometer. The Myriapod Colon status kit (Diatech Pharmacogenetics, Jesi, Italy) was used to evaluate the mutational status of *KRAS* (exons 2, 3 and 4), *NRAS* (exons 2, 3 and 4) and *BRAF* (exon 15). This analysis is based on MALDI-TOF Mass Spectrometry technology associated with the Single Base Extension. The panel consisted of 8 multiplexes for sample, for each reaction 5–50 ng of DNA was used and analyzed the *RAS* and *BRAF* gene. The analysis was performed according to manufacturer’s instructions. The results were analyzed using the analysis software of instrument. The sensitivity allowed the detection of mutation was 2.5%.

### 2.5. Additional Exploratory Analyses

Some samples were analyzed with Idylla^TM^ platform, a fully automated molecular system (Biocartis, Mechelen, Belgium) [23]. This technology combines sample preparation with PCR thermocycling and fluorescence detection of target sequences.

We used an aliquot of the same samples analyzed by ddPCR. These were then processed and frozen at −80°C until they were used for Idylla analysis. One microliter of plasma sample was directly dispensed into the disposable cartridge test and the experiment started. Within approximately 2.5 h, with a hands-on time of less than 2 min, Idylla^TM^ returns a real-time and reliable genotyping in patients with mCRC [26].

The sensitivity cut-off for the DNA detection assay was set at the lower limit of 1% mutant alleles.

### 2.6. uctDNA Analyses

Urine samples taken for ddPCR analysis were subjected to pre-analytical processes, centrifuged to remove debris and contaminants. Urine sample containers were pre-filled with 10 mL EDTA (0.5 M, pH 8.0) and approximately 30 mL of urine was collected for each patient. Subsequently, urine samples were transferred and preserved into cfree(f) DNA-BCT^®^ tubes (Streck, Omaha, NE, USA). In this study, such containers were used to ensure comparability between ctDNA extracted from plasma and urine samples, as the same ddPCR method was used for the analysis.

The cell fraction was removed by centrifugation (20 min at 2000× *g* at +4 °C) and the supernatant was carefully transferred into a 15 mL tube. A second centrifugation was performed at 16,000× *g* at +4 °C to remove any cellular residues; 4 mL of processed urine was recovered for DNA extraction. The ctDNA was extracted using the QIAamp^®^ ccfDNA Mini kit (Qiagen GmbH, Hilden, Germany) according to the manufacturer’s protocol. The isolated ctDNA was stored at −20 °C until analysis.

### 2.7. Statistical Analyses

Variables were described using mean and median when continuous, and percentage when categorical.

To compare turnaround time (T1) of LB vs. STB (T2)—our primary endpoint—we used the Mann–Whitney U test.

To evaluate the accuracy of plasma-based *RAS/BRAF* testing we evaluated sensitivity, specificity, negative and positive predictive values. Sensitivity was defined as the ratio between true positive (TP) cases and the sum of TP and false negative (FN) cases, specificity as the ratio between true negative (TN) cases and the sum of TN and false positive (FP) cases. Negative predictive value was estimated as the ratio between TN cases and the sum of TN and FN, positive predictive value as the ratio between TP cases and the sum of TP and FP ones.

## 3. Results

### 3.1. Patient Characteristics

From September 2018 to March 2019, a total of 33 consecutive external patients with histologically confirmed mCRC were included: 21 males and 12 females having a mean age of 60 years (range: 39–91 years).

Patient baseline characteristics, and number and location of metastasis are summarized in Table 1 and Table 2.

Primary tumor was not resected prior to liquid biopsy in 9/33 patients (27%) and located in rectum, left and right side in 18%, 49% and 33% of cases, respectively.

Regarding the metastatic sites, liver was the most frequent site of metastasis (22/67, 67%), followed by lymph-nodes (17/33, 51.5%).

Considering the previous treatments, a total of 24 patients (73%) had newly diagnosed disease and were naive to treatment. With respect to treated patients, 15% (5/33) received at least two lines of prior chemotherapy.

All patients were referred to our hub center for a second opinion to evaluate the systemic treatment initiation after that STB *RAS/BRAF* testing was performed by our center as we usually do at the same time as the first oncological visit.

### 3.2. RAS/BRAF Mutational Status Analysis from Tissue and Plasma

#### 3.2.1. Turnaround Times between STB Analysis and Plasma LB

The T1 mean was 7 days (d) with the same median value (range: 2–12 d), whereas the LAB-time mean was 4 d with a median value of 5 d (range: 1–10 d).

The T2 mean was 22 d with a median value of 17 d (range: 7–65 d), whereas the mean and median of PAT-time were 6 d (range: 1–18).

T1 resulted significantly shorter when compared to T2 using Mann–Whitney U test (*p* < 0.00001). The comparison of T1 vs. T2 results (primary endpoint) is shown in Figure 1. 

#### 3.2.2. Concordance between STB Analysis and Plasma LB

Regarding the STB analysis, in 3 patients out of 33 *RAS/BRAF* genotyping was not carried out because one specimen was out of stock and two other ones resulted unretrieved at the data-lock point of the present study. Thus, we excluded these three patients from the STB analyses in terms of concordance with LB testing on plasma.

Among 30 patients, 15 (50%) harbored *RAS* mutations in STB genotyping and 14 (42%) in plasma-based analysis. V600E *BRAF* mutation was detected in STB and LB testing in five subjects. A total of 10 (33%) tumoral tissue specimens resulted all wild type, whereas no *RAS/BRAF* mutations were found in 14 (42%) plasma samples (Table 3).

The *RAS/BRAF* mutational status determined by ddPCR from plasma corresponded to the data STB analysis in 25/30 patients (83% overall percent agreement).

Considering tissue analysis as gold-standard for tumor genotyping, we calculated TP, FP, TN and FN results obtained from plasma-based LB. TP and FP cases were 16/30 and 1/30, respectively; FN were 4/30 and TN were 9/30. These data resulted in a 90% specificity and 94% PPV. Conversely, sensitivity and NPV were 80% and 69%, respectively.

The concordance of *RAS/BRAF* status between LB and STB analysis from each patient (secondary endpoint) is summarized in Table 4.

#### 3.2.3. Turnaround Times and Concordance between STB Analysis and Plasma KRAS Testing by Idylla^TM^ Method

A total of four plasma samples from *KRAS* mutated patients based on STB analyses were tested by means of Idylla^TM^ platform technique [23]. We were able to confirm the turnaround time of 120 min in all (100%) analyzed samples. The Idylla^TM^ ctDNA assays confirmed STB results in three out four *KRAS* mutations (i.e., G13D, G12A and G12V). Interestingly, the only one case wild type at Idylla^TM^ was reported to be wild type also at ddPCR.

### 3.3. Description of False Results from RAS/BRAF Mutational Status Analysis by Plasma ddPCR

False results subgroup (N = 5) included one FP and four FN cases in comparison to the gold standard STB genotyping.

This subgroup was analyzed in depth taking into consideration major clinical features (i.e., age, CEA level) and baseline characteristics of primary tumor (i.e., single/multiple primary tumor, primary tumor location and resection, mucinous histology). We also investigated major characteristics of metastatic disease at time of LB and we reviewed the CT scan carried out closest to the blood extraction to calculate total tumor volume (cm^3^), as a good surrogate of tumor burden.

Overall, FN cases had lower total tumor volume (8.5 cm^3^ vs. 52.6 cm^3^) and lower number of total metastatic lesions (4.2 vs. 9.2) compared to those in true cases. All these data are summarized in Figure 2.

The only FP case was patient number 20, in which LB identified a *KRAS* mutation with a MAF of 120 copies/mL, whereas it was identified as all wild type by tumoral STB genotyping.

### 3.4. MAF Analysis

For the 19 patients with detectable plasma *RAS* (14) or *BRAF* (5) mutations in LB analysis, the mean amount was 47.70 copies/mL (range: 80–628000 copies/mL). In particular, *RAS* had a mean value of 58.38 copies/mL (range: 80–628000 copies/mL) and *BRAF* had a mean value of 1400 copies/mL (range: 600–3000 copies/mL).

Given the possibility that the amount of circulating mutant alleles is related to the overall tumor burden or extent of metastatic invasion, the copies/mL of mutant alleles in plasma were compared to the total tumor volume. We observed that the samples with the highest number of copies/mL correlated to the largest total tumor volume.

### 3.5. RAS/BRAF Mutational Status Analysis from Tissue and ucfDNA

A total number of 19 consecutive mCRC patients with known *KRAS* mutation status were included. Baseline patients and primary tumor characteristics, metastatic disease features at the time of urine collection and previous treatments information are summarized in Table 5.

*KRAS* mutation was detected in 12 (63%) STB analysis and in 10 (53%) ucfDNA samples. All seven patients *KRAS* wild type on tumor biopsy were wild type also on urinary LB (Table 6).

Overall, concordance of *KRAS* mutational status between STB and urine-based LB testing was 89.5% (17/19 cases).

Accuracy of *KRAS* mutation detection on ucfDNA compared to STB is described in the following values: 83% of sensitivity, 100% specificity, with a positive predictive value and negative one of 100% and 78%, respectively.

As well as for FN cases identified on plasma-based LB, we analyzed in depth the two FN results from urine-based LB taking into consideration clinical and pathological features. We observed that the first one underwent primary tumor resection before urine collection and had a mucinous histology; the second one presented with only extra-liver metastases (brain and lung); both were treatment-naive.

## 4. Discussion

The advent of the targeted therapy meant a step further in the direction to achieve the selection of the right therapy, to the right patients, at the right time. Concerning the management of mCRC patients, the accurate prescription of anti-EGFR agents is of high clinical importance. Thus, current guidelines recommend expanded *RAS* analysis in order to identify patients for anti-EGFR therapy more precisely [15,27]. In particular, at initial diagnosis the timing of molecular testing results is relevant for first-line treatment. Despite the great steps forward [28,29,30], in two recent studies of mCRC patients evaluated for first-line therapy in the United States of America, approximately 30% of subjects failed to be tested on *RAS* mutational status [31,32]. In the real-world data reported by Sangaré et al. [32] and performed in 255 American public hospitals on a sample of over 17,000 patients, the *RAS* testing turnaround time was >15 days in 80% of study population [32]. These data are consistent with a European report [30] according to which the turnaround time of *RAS* testing can be extended beyond 10 days for external samples. These findings were confirmed by a survey of European physicians where the *RAS* testing turnaround time and the tissue unavailability resulted the most frequent factors cited for treating mCRC patients with unknown *RAS* status [33].

Moreover, external quality assessment surveys found that there is still great variability between country and country in the methods and times of *RAS* analysis, and that the latter expand especially in the cases of samples to be recovered [34,35]. All these factors contribute to hinder a wider realization of tailored therapy. An accurate blood-based *RAS/BRAF* mutation assay with fast turnaround time would help circumvent these hurdles.

To our knowledge, the present analysis is one of the very few studies that shows in a daily clinical routine the usefulness of detecting *RAS/BRAF* point mutations by ctDNA providing a head-to-head comparison in terms of turnaround time between liquid (i.e., plasma and urine) and solid testing.

The turnaround time of plasma analysis—the primary endpoint of our study—resulted faster compared to tumor-tissue analysis (STB). The STB tissue-based method had a mean and median turnaround time (T2) of 22 d and 17 d, respectively, whereas the plasma-based (LB) method had both mean and median turnaround time of 7 d (i.e., T1). Notably, we calculated T2 as the interval between the tissue-based results and the request for specimen recovery.

All tumor tissue samples were analyzed at the department of pathology of University of Padua but they were all external specimens, made and stored at various and distant centers. This is why the mean and the median T2 resulted superior to that observed in a previous study (mean 13 d, median 11 d) [35], which proposed a clinical validation of the *RAS/BRAF* mutations’ detection from ctDNA in mCRC patients. In the mentioned analysis a large quote (>90%) of tumor tissue genotyping was carried out in the clinical center where the FFPE specimens were made and stored [36].

In our clinical series, the main reason behind such a wide difference between the two turn-around times (T1 and T2) is the time needed to recover external tumoral tissue specimens. As matter of fact, the interval mean between the sample request and its receipt at our center was 17.2 d (median value was 12 d, range 1–65 d), whereas the interval between the tissue-sample entry to our pathology unit and STB result (i.e., PAT-time) was 6 d only, as described above.

A necessary further step toward implementing blood- or urine-based *RAS/BRAF* testing in the management of mCRC patients is to demonstrate concordance between ctDNA and tissue mutational status analysis, and to evaluate the accuracy of LB. Indeed, the secondary endpoints of our study were to analyze the concordance and the sensitivity, specificity and predictive values of ctDNA genotyping.

Our results showed a very high overall concordance, close to 90% (83% for plasma- and 89% for urine-based ddPCR), in comparison to the gold-standard tumor tissue analysis. These data are consistent with several retrospective and prospective reports [21,37,38,39,40], whereas in most of them liquid and solid biopsies were almost concomitant. Conversely, in our series the tumoral tissue sampling was concurrent with the blood or urine sampling in a very small cases only: over 85% of *RAS/BRAF* genotyping was performed from tumoral tissue specimens collected at least one year before LB execution. For this reason, a similar value of concordance as emerged in the present study is even more noteworthy.

In our cohort, the plasma-based ddPCR analysis provided five false results (1 FP + 4 FN). Among them, a total of one and three patients did not harbor *BRAF* and *KRAS* mutation, respectively, that were detected in tissue samples.

Lack of *RAS* or *BRAF* mutations in plasma should be investigated because of the possible detrimental interaction of anti-EGFR agents in *RAS* mutant patients. FN results may be attributed to emerging biological factors that affect ctDNA shed.

Commonly, the molecular landscape can be modified by chemotherapeutic and targeted agents in these tumors revealing an increasing number of acquired *KRAS* mutations that are notoriously correlated with secondary resistance to EGFR blockade [41,42]. However, the impact on the molecular profile derived from other therapies such as antiangiogenics or cytostatic agents deserves to be further investigated [43,44].

Recently, Tie et al. [45] observed changes in ctDNA shed for mCRC patients during the chemotherapy administration, with significant reduction in ctDNA values reported before cycle 2 in 41 of the 48 patients with concordant mutant samples in ctDNA and tissue. This could explain the lack of *RAS* mutation detection in patients recently exposed to chemotherapy, but it does not explain our four FN cases which were all chemo-naive.

Among the FN subgroup, the primary tumor was resected in two patients (i.e., patient number 2 and number 23), which presented metachronous lung metastasis at the time of LB. Primary tumor tissue genotyping resulted *KRAS* G12S and *BRAF* V600E mutated, respectively, whereas no *RAS* neither *BRAF* mutations were detected by cfDNA analysis. This discordance could only be partially attributed to the molecular heterogeneity between primary and metastatic lesions. An increasing number of studies considering inter-tumor heterogeneity (between primary tumors and metastases in the same patient) have demonstrated mutational discordance in 3.6–32.4% of cases [36,46,47].

Another and more likely explanation of this discordance is that detection of *RAS/BRAF* mutation in ctDNA could be negatively affected by the low tumor load. An increasing number of analyses pointed out that the frequency of circulating mutant alleles is related to overall tumor burden or extent of metastatic invasion [48,49,50]. Indeed, when considering the copies/mL of the mutated subgroup, we observed a trend towards lower *RAS/BRAF* plasma MAFs in patients with lower tumor burden (defined as total tumor volume). Overall, false cases had lower total tumor volume (8.5 cm^3^ vs. 52.6 cm^3^) and lower number of total metastatic lesions (4.2 vs. 9.2) compared to that in true cases.

In addition to the tumor burden, the site of metastasis (i.e., lung) could negatively impact copies/mL in patient number 2 and number 23. A large number of analyses demonstrated that patients with liver involvement has higher MAFs compared with those without liver metastasis [39].

We could draw other interesting insights from analyzing copies/mL due to the relatively large cohort size of patients with detectable *RAS* or *BRAF* mutations in our study (N = 19/30). A widely large range (i.e., 80–628000 copy/mL) was showed, suggesting very high interindividual heterogeneity. This is consistent with the amount distribution reported in other studies [48,49,50]. As a matter of fact, a plethora of clinical and pathological characteristics could influence ctDNA shed such as the site of metastasis (peritoneum, lung), mucinous primary tumor histology and previous treatments. In the effort to better understand the biology of ctDNA as well as to explain plasma/tissue discrepancies in our cohort, we evaluated all these clinical and pathological features: data from plasma-based analysis suggested that tumor burden rather than intrinsic biological characteristics of tumor may impact ctDNA release. In contrast, our exploratory analysis on uctDNA would seem to strengthen the role of biological traits: out of the two FN results, the first one had a mucinous histology and the second one presented with only extra-liver metastases (brain and lung).

However, tumor heterogeneity and variable DNA shed fail to fully account for the recurring reports of inaccurate ctDNA genotyping reported in literature [13,15,17,36,48,51,52]. Similarly, the only FP case (patient number 20) of our clinical series could bring into play other issues. He was identified as all wild type on STB genotyping, whereas LB detected a *RAS* mutation with a low MAF (120 copies/mL). At the time of blood sampling, he presented with sepsis and had undergone amputation of all toes due to a concomitant acute ischemia. We repeated plasma-based *RAS* analysis after 2 weeks of antibiotic therapy and any *RAS* mutation was no longer found.

Since a relevant percentage of cfDNA comes from peripheral blood cells (PBC) [48,53,54], we supposed that somatic mutations within non-malignant hematopoietic cells, known clonal hematopoiesis [53,54], could explain this FP result. Recently, rare *KRAS* mutations detected in cfDNA were demonstrated derived from clonal hematopoiesis, not from tumor [54]. The clearest way to overcome this challenge could have been paired genotyping of both plasma cfDNA and PBC DNA. Unfortunately, PBC were discarded as usual when spinning plasma, so this analysis was not carried out.

Regarding our results of uctDNA analysis, the reported agreement, specificity and sensitivity (89.5%, 100% and 83%, respectively) are satisfactory considering the higher risk of contamination and degradation of urine sample and the absence of validated pre-analytic and analytic protocols. They are comparable to those reported in previous studies where the agreement fluctuates between 70% and 90%, while the sensitivity is very variable and often suboptimal (63–93%) also due to the inadequate volume of urine samples [55,56,57,58,59,60]. However, further studies on a larger number of patients are needed to confirm these promising results and explore the potential of uctDNA which could in some cases replace plasma biopsy due to its ease of execution and not requiring nursing procedures. To date, liquid urinary biopsy has been mainly investigated in lung and urinary tract tumors [55,57,60].

## 5. Conclusions

Our results in terms of faster turnaround time, very high concordance and accuracy are three key points to implementing *RAS/BRAF* blood-based analysis in the day-by-day care with mCRC patients, in particular in the urgent situation to define critical decision points for the anti-EGFRs administration.

CtDNA in plasma represents a quick, feasible minimally invasive source of DNA alternative to tissue biopsy, that offer a real-time assessment of the cancer mutation status thanks to its repeatability during the patient’s history. However, the analysis of ctDNA in plasma still needs an extensive clinical description, as confirmed by our data. In the present work, the plasma-based method exhibited 80% sensitivity and 69% NPV, in accordance with several previous studies [13,36,51,52]. As a matter of fact, the major technical challenges in testing ctDNA are the low fraction of the mutated allele, which is highly variable in different patients, and the broad dynamic range. Consequently, the most recent efforts are focusing on developing highly sensitive methodologies such as ddPCR and Idylla^TM^ testing [23,26,61,62]

In our opinion, ddPCR is the most useful procedure in a daily clinical routine compared to Next Generation Sequencing and Idylla^TM^, due to low costs and time-saving methodologies. Additionally, ddPCR allows a quantitative assessment of the fractional abundance of the mutation.

Despite the great value of the results presented, there are few limitations to our study. First of all, longitudinal blood extractions were not carried out in order to monitor dynamic mutational load, so our conclusions should be cautiously interpreted, in particular regarding correlation between MAFs distribution and specific clinical and pathological characteristics. Additionally, a progression free survival (PFS) analysis was not performed in order to evaluate potential impact in PFS of various MAF levels because survival data are still immature in our study [50,63].

The undoubted advantages of genotyping with cfDNA are now recognized by international guidelines and regulatory agencies, in particular, the FDA approved the blood-based detection of EGFR mutations in NSCLC to identify eligible patients for tyrosine kinase inhibitors. Similarly, in the future, ctDNA may be used in the field of mCRC to help select *RAS* wild-type patients suitable for anti-EGFR agents. There is a growing body of evidence to improve the broader development of ctDNA as a biomarker in several settings of CRC [64,65,66,67,68,69,70,71]. Indeed, increasing efforts of translational research are now directed in the long way to introduce the ctDNA analysis in carefully designed large prospective trials [67,68,70].

Nevertheless, several challenges need to be overcome before this potential biomarker is introduced in our daily clinical practice with mCRC patients: First of all, pre-analytical issues such as the standardization of sample collection, processing and storage [72,73,74]. Moreover, there is still no consensus on the interpretation of results, in particular regarding the number of mutant droplets needed to interpret results as positive. There are many questions still answered. For example, should the cut-off of mutant droplets be lower in the early disease setting despite the resulting risk of too many false positives? Recent studies have tracked an increasing number of mutations in order to pick up relapse more reliably [75,76,77]. How much does it impact in terms of cost implications? Finally, could ctDNA also appear to be an independent prognostic biomarker in addition to its predictive capability, as recently suggested by El Massaoudi et al. [50]?

Despite these challenges, we shone a light on the first potential use of ctDNA in daily clinical practice to guide therapy in mCRC patients, in particular when decision on first-line therapy is urgent and tissue recovery from external centers may require a long time. Moreover, we expect that our increasing experience genotyping and interpreting ctDNA will allow to reach the international consensus that is still lacking and will address the other aforementioned open issues, finally introducing ctDNA in day-to-day clinical practice for many CRC management scenarios.

## Figures and Tables

**Figure 1 cancers-13-05128-f001:**
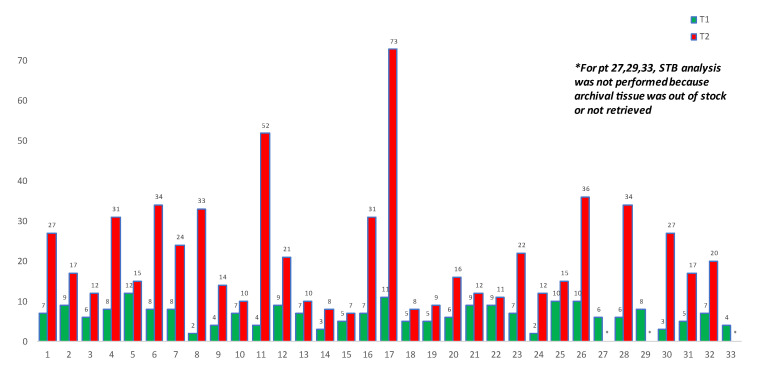
Comparison of time to LB (T1) vs. STB (T2) results.

**Figure 2 cancers-13-05128-f002:**
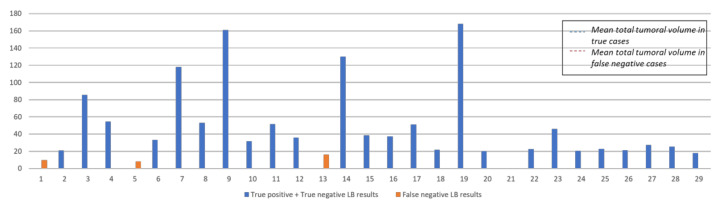
Comparison of false negative vs. true results of LB in terms of tumor volume.

**Table 1 cancers-13-05128-t001:** Major clinical features and baseline characteristics of primary tumors at time of LB on plasma.

Characteristic		Number (%)
Age (y.o.)	mean	60
	range	39–91
Sex	Male	21(64%)
	Female	12(36%)
CEA level >5 ng/mL	Yes	19 (58%)
	No	14 (42%)
Primary tumor single lesion	Yes	29 (88%)
	No	4 (12%)
Primary tumor location	right	11 (33%)
	left	16 (49%)
	rectum	6 (18%)
Primary tumor resected	Yes	24 (73%)
	No	9 (27%)
Mucinous histology	Yes	7 (21%)
	No	21 (64%)
	NA	5 (15%)

**Table 2 cancers-13-05128-t002:** Major characteristics of metastatic disease at time of LB on plasma.

Characteristic		Number (%)
Number of metastatic sites	single	13 (39%)
	multiple	20 (61%)
Liver metastasis	Yes	22 (67%)
	No	11 (33%)
Number of total metastatic lesions	mean	9.2
	range	1–25
Total tumor volume (cm^3^)	mean	158
	range	0.9–248
Number of CT lines prior to LB	0	24 (73%)
	1	4 (12%)
	2	4 (12%)
	>2	1 (3%)
Anti-angiogenics prior to LB	Yes	9 (27%)
	No	24 (73%)

**Table 3 cancers-13-05128-t003:** Description of LB and STB results in terms of *RAS/BRAF* status.

RAS/BRAF Status	LB	STB
KRAS mutated	13/33 (39%)	14/33 (43%)
NRAS mutated	1/33 (3%)	1/33 (3%)
BRAF V600E mutated	5/33 (15%)	5/33 (15%)
All wild-type	14/33 (43%)	10/33 (30%)
NA	0 (0%)	3/33 (9%)

**Table 4 cancers-13-05128-t004:** Accuracy of plasma-based LB vs. STB analysis.

	RAS, BRAF Mutated on STB (*n*)	All Wild-Type on STB (*n*)
**LB +**	True Positive (16)	False Positive (1)
**LB −**	False Negative (4)	True Negative (9)

**Table 5 cancers-13-05128-t005:** Major characteristics of patients and primary and metastatic disease at time of LB on urine.

Characteristic		Patients (%) *n* = 19
Age mean (Y)		63 (42–83)
CEA level >5 ng/ml		13 (68)
Primary tumor single lesion	single	18 (95)
	multiple	1 (5)
Primary tumor location	right	5 (26)
	left	8 (42)
	rectum	6 (32)
Primary tumor resected	Yes	12 (63)
	No	7 (37)
Mucinous histology	Yes	4 (21)
	No	9 (47)
	NA	6 (32)
Number of metastatic sites	single	9 (47)
	multiple	10 (53)
Site of metastasis	Liver	11 (58)
	Non-liver	8 (42)
Number of total metastatic lesions (median)		15.2
Total tumor volume (cm3) (median)		190
Number of CT lines prior to LB	0	12 (63)
	1	4 (21)
	2	3 (16)
	>2	0

**Table 6 cancers-13-05128-t006:** Accuracy of urine-based LB (uLB) vs. STB analysis.

	RAS, BRAF Mutated on STB (*n*)	All Wild-Type on STB (*n*)
**uLB+**	True Positive (10)	False Positive (0)
**uLB−**	False Negative (2)	True Negative (7)

## Data Availability

The data presented in our study are available on request from the corresponding author. The data are not publicly available due to our internal policy in terms of privacy restrictions.

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
