# Peer review of "A Real-World Application of Liquid Biopsy in Metastatic Colorectal Cancer: The Poseidon Study"

_cancers, 2021, doi:10.3390/cancers13205128_

Round 1

Reviewer 1 Report

Congratulations to the authors for the interesting work. The paper is thoroughly developed, also with respect to the limitations of the trial.

I have just a couple of curiosities:
- According to the Italian guidelines on the observational studies, the trial is borderline between an observational  and an interventional non-pharmacological one. Did any Ethical Committee member suggest you to present it conforming to the latter definition?

- Has the trial been registered on clinicaltrials.gov? If yes, I advise you to include its ID number in the article.

- The EC approved the study in 2015, but the enrolment opened in September 2018 only. What is the reason of such a major time lag?

Reviewer 2 Report

Dear Authors,

The subject of the work is interesting and now widely discussed. The ctDNA testing is becoming popular, so it is important that the centers exchange their own experiences, especially regarding the methodology of material collection, transport and methods of detection of mutations in ctDNA. Standard KRAS/NRAS/BRAF testing in tumor biopsies has some drawbacks (eg. insufficient/not accessible tumor tissue, bad quality) and only tumor fragment is tested. Analysis of circulating tumor DNA (ctDNA) released from malignancy cells seem an alternative method, also for monitoring the therapy response.

I evaluate the work positively, but I think that some methodical aspects should be described in more details, and the description of molecular analysis should be extended. Moreover, precise information on how the material was collected (were Streck tubes used in all cases?) is missing; was the plasma always frozen and sent to the external laboratories (what was the situation with the research on Idylla?);  were  ddPCR determinations performed in triplicate per case (especially important for ctDNA negative cases, and tissue positive cases)? In addition, I suggest that the mutation name should be given (if the test used allowed for the identification of individual genetic changes). The method used to identify the mutations in the tissue allowed for the determination of mutations in three KRAS exons, whereas the ddPCR method only in one (codons 12 and 13, has it been verified in tissue which exact mutation were detected? May this be the reason of some discrepancies?). Why not all 30 cases were tested with the Idylla test? This would be valuable for the comparison of methods for detecting mutations in ctDNA (Idylla has the ability to identify mutations in three exons of the KRAS gene, just like the tissue reference method used in the study). The study group itself contains very low number of cases.

Best regards,

Reviewer 3 Report

Procaccio et al. present a study evaluating the utility of liquid biopsy for the detection of BRAF/RAS mutations. Though interesting, there are a few concerns that warrant  consideration. Please also review the manuscript for spelling and grammar.

Introduction

Line 83 – spelling error for non-small cell lung cancer

Line 83 – I don’t understand the comparison between NSCLC and colorectal cancer for tumor tissue availability. That clause can be removed.

Lines 84-87 – Though tumor tissue is more abundant, it is still invasive to obtain. I think this fact warrants discussion.

Lines 89-95 – I’m confused by this paragraph. I think this paragraph is supposed to set up the argument for the manuscript, but it is not worded clearly. Please rewrite and clarify your argument.

Why focus on metastatic CRC rather than non-metastatic? Does this have to do with the abundance of ctDNA available in metastatic versus non-metastatic? I believe this is an important point to clarify.

Lines 99-107 – Combine these two paragraphs.

As the purpose of the paper is to compare the two methods (STB v. LB), it’s necessary to discuss the limitations to STB and why a different method is needed. This information should be added to the Introduction.

Methods

Lines 115-117 – The primary objective is out of place in the Methods. It is better suited for the Introduction.

Lines 115-128 – Though I understand that the authors are including a lot of measure in the manuscript, there is an overwhelming number of abbreviations. Please limit some of the abbreviations.

Lines 147-149 – This information should be moved to the next subsection.

Line 154 – Change DdPCR to ddPCR

Use of Mann-Whitney U tests (and other statistical tests) for comparisons is in the Abstract, but this information is missing from the Methods. This needs to be added.

Results

Table 1 – The text states 60 years for the median, though the table has 60 years for the mean. Are the mean and median the same? If not, please edit accordingly.

Table 1 – Include male/female in the table.

Much of the Results are presented as single sentence paragraphs. This needs to be edited for proper sentence/paragraph structure.

Figure 1 – I am unable to read the legend in the upper right corner. Please make this bigger for the reader.

Line 238 – Change ‘showed’ to ‘shown’

Table 3 – Please include statistics in the table.

Line 258-259 – Please add the percent concordance between LB and STB in the text.

Line 267 – Change ddPRC to ddPCR

Section 3.2.3 – Concordance is in the subsection title, but not in the text. What is the purpose of including the turnaround time in the results? Is there a comparison to LB?

Figure 2 – This information would be better suited for a table rather than a figure. Also, as comparisons are given in the text, this information should be in the table, along with statistics.

Lines 287-288 – numbers can be round to two digits.

Lines 286-289 – Can statistics be presented?

Lines 290-293 – This information would be really great for a figure, showing the comparison of alleles to tumor volume.

Table 5 – For age, is mean or median being presented?

How does Table 5 differ from Tables 1 & 2? If Table 5 is similar to Tables 1 & 2 (except for urine), perhaps combine and make the distinctions clear.

What is the concordance between LB and uLB?

Discussion

The first part of the Discussion should be a summary of the study, prior to jumping into the literature.

Lines 347-353 – I don’t understand why having specimens from different sites resulted in improved mean and median T2. I would think that using external specimens would result in longer means and medians. Please describe further.

The Discussion needs to be edited down and made more concise.

Conclusions

Line 454 – Getting plasma from patients isn’t non-invasive, but minimally invasive.

I think that it could be useful to discuss the potential differences in MAF by cancer stage. Do the authors think that MAFs will be reliably detected in early stage disease? What would this mean for detection and treatment?
